

# Speciation of OH reactivity above the canopy of an isoprene-dominated forest

J. Kaiser[1], K. M. Skog[1], K. Baumann[2], S. B. Bertman[3], S. B. Brown[4,5], W. H. Brune[6], J. D. Crounse[7], J. A. de Gouw[4,5,8], E. S. Edgerton[2], P. A. Feiner[6], A. H. Goldstein[9,10], A. Koss[4,8], P. K. Misztal[9], T. B. Nguyen[7], K. F. Olson[9], J. M. St. Clair[7], A. P. Teng[7], S. Toma[3], P. O. Wennberg[7,11], R. J. Wild[4,8], L. Zhang[6], and F. N. Keutsch[1]

[1]Department of Chemistry, University of Wisconsin-Madison, Madison, WI, USA
[2]Atmospheric Research & Analysis Inc, Cary, NC, USA
[3]Department of Chemistry, Western Michigan University, Kalamazoo, MI, USA
[4]Chemical Sciences Division, NOAA Earth System Research Laboratory, Boulder, CO, USA
[5]Department of Chemistry, University of Colorado, Boulder, CO, USA
[6]Department of Meteorology, Pennsylvania State University, University Park, PA, USA
[7]Division of Geological and Planetary Sciences, California Institute of Technology, Pasadena, USA
[8]Cooperative Institute for Research in Environmental Sciences, University of Colorado Boulder, Boulder, CO, USA
[9]Department of Environmental Science, Policy, and Management, University of California, Berkeley, California, USA
[10]Department of Civil and Environmental Engineering, University of California, Berkeley, California, USA
[11]Division of Engineering and Applied Science, California Institute of Technology, Pasadena, USA
[12]School of Engineering and Applied Sciences and Department of Chemistry and Chemical Biology, Harvard University, Cambridge, Massachusetts, USA

*Correspondence to*: J. Kaiser (jen.b.kaiser@gmail.com)

**Abstract.** Measurements of OH reactivity, the inverse lifetime of the OH-radical, can provide a top-down estimate of the total amount of reactive carbon in an airmass. Because OH reactivity is tied to the $RO_2$ production rate, the absolute value of OH reactivity has direct implications for ozone production. Additionally, as molecular structure determines volatility, the speciation of reactivity affects the production of secondary organic aerosol (SOA). Several studies have focused on the agreement of measured and calculated or modeled OH reactivity above and within the canopy of isoprene-dominated forests, as well as the relative contributions of volatile organic compounds (VOCs) and oxidized VOCs (OVOCs). Drawing definitive conclusions about the identity of the missing OH reactivity has been limited by the availability of VOC and OVOC measurements. In this work, using a comprehensive measurement suite, we examine the measured and modeled OH reactivity above an isoprene-dominated forest in the South East United States during the 2013 Southern Oxidant and Aerosol Study (SOAS) field campaign. We find good agreement between measured and modeled OH reactivity, with the largest contribution consistently coming from primary biogenic emissions. In contrast, there are small but significant discrepancies in the increase in OH reactivity per isoprene. As the model typically overestimates OVOCs, we do not attribute this discrepancy to unmeasured oxidation products. Instead, we suggest that unmeasured primary emissions may influence the OH reactivity at this site.





# 1 Introduction

Biogenic emissions of volatile organic compounds (VOCs) constitute the largest source of reactive carbon in the atmosphere (Guenther et al., 2012). During the daytime, oxidation of VOCs by the OH radical can drive the formation of secondary pollutants. Under high $NO_X$ ($NO + NO_2$) conditions, the peroxy radical ($RO_2$) generated from VOC oxidation can convert

NO to $NO_2$, which ultimately photolyzes to form ozone. Additionally, oxidized VOCs (OVOCs) are typically less volatile than their precursors, and can contribute to the formation of secondary organic aerosol (SOA). Therefore, it is important to understand the total VOC + OH reaction rate and the fate of the resultant OVOC to understand the formation of secondary pollutants such as tropospheric ozone and SOA.

While measuring every VOC and oxidation product OVOCs is not feasible, measurement of OH reactivity (the loss rate of

the OH radical divided by the OH concentration) provides an alternative to the bottom-up molecular approach (Kovacs and Brune, 2001). The absolute value of OH reactivity can be used as a top-down estimate of the total amount of reactive carbon in an airmass, or the $RO_2$ production rate. The speciation of the reactivity carries air-quality relevant implications as SOA yield is directly tied to molecular properties such as volatility, hygroscopicity, viscosity, and condensed-phase reactivity.

One half of the annual non-methane VOC emissions is in the form of isoprene ($C_5H_8$), making it the dominant biogenic VOC

globally (Guenther et al., 2012). Due to isoprene's abundance and high reactivity, the chemistry of isoprene and its resulting oxidation products have been the focus of numerous field studies. OH reactivity has been examined in four isoprene-dominated forests, with some studies suggesting missing primary emissions or missing OVOCs, and others finding good agreement between measurements and calculations.

For example, using measured VOCs and OVOCs, Di Carlo et al. (2004) could account for only 50% of the OH reactivity

measured above the canopy of a deciduous forest in northern Michigan in the summer of 2000. As OVOCs calculated by a model did not significantly increase calculated OH reactivity, and as the missing reactivity fit a terpenoid-like emission profile, unmeasured terpene emissions were cited as a large source of reactive carbon in this environment. However, later measurements of monoterpenes and sesquiterpenes at this site suggested that only ~20% of the missing reactivity could be attributed to these primary VOCs, leaving a total of ~40% unaccounted measured reactivity (Kim et al., 2009). Additionally,

Kim et al. (2011) found that measurements and calculations of OH reactivity in branch enclosures of isoprene-emitting trees at the same site were in good agreement. They calculated ~8% of the missing reactivity could be attributed to unmeasured isoprene oxidation products. Recently, from 2009 measurements at the same site, Hansen et al. (2014) found that isoprene accounted for 60-70% of afternoon OH reactivity both within and above the forest canopy. Because in-canopy OH reactivity calculations and measurements were in good agreement, the authors concluded there are unlikely to be unmeasured VOCs at

this site. However, above-canopy comparisons show a large missing fraction of reactivity, suggesting unmeasured oxidation products may contribute at longer processing times.

Zannoni et al. (2015) examined OH-reactivity both within and above a downy oaks forest in the Mediterranean south east of France. Measured and calculated OH reactivity were in good agreement at both heights during the daytime, with isoprene





contributing 83% within the canopy and 74% above the canopy. However, more than 50% of nighttime reactivity was missing on a subset of days. The authors conclude that unmeasured, higher-generation isoprene oxidation products are part of the nighttime discrepancy, as well OVOCs from the reactions of large, non-isoprene biogenic VOCs and ozone.

In contrast, in a tropical rainforest on Borneo, unmeasured isoprene-derived OVOCs were a more dominant contribution to the observed reactivity than isoprene itself, at nearly 50% (Edwards et al., 2013). OH reactivity measured from a clearing atop a hill surrounded by forest was significantly underestimated by a model (~60% at noon). The authors concluded missing primary emissions were unlikely to contribute significantly to OH reactivity, and an underrepresentation of secondary multifunctional OVOCs is a likely source of discrepancies.

Finally, in-canopy OH reactivity measured in the tropical rainforest of Suriname could not be reached by summing the contributions from measured isoprene, methyl-vinyl ketone, methacrolein, acetone, and acetaldehyde. The authors called for more comprehensive measurement suite to investigate the large discrepancy (65%) (Sinha et al., 2008).

Each forest used in these studies is composed of a unique species of trees, potentially leading to different relative biogenic VOC emissions. Furthermore, different meteorological conditions and canopy structures may lead to different processing times and resultant contributions of isoprene-derived OVOCs. While these differences may make the above studies difficult to generalize, they all address an underlying question: are isoprene-derived OVOCs a substantial source of missing reactive carbon? If so, after what degree of processing? What is the impact of $NO_X$?

To assess the contribution of unmeasured oxidation products, ideally, one would explicitly model all isoprene OVOCs and include modeled species in the summation. Additionally, several OVOC measurements would be available to test the reliability of model concentrations. This sort of analysis has been performed in a chamber study of the oxidation of isoprene (Nölscher et al., 2014), but as initial concentrations of reactants were orders of magnitude greater than those found in the atmosphere, and as physical processes such as deposition onto plant surfaces are not captured in chamber studies, chamber experiments may not capture the behavior of OH reactivity observed in a forest.

Of the above field studies, several rely only on the concentration of measured species in the calculation of OH reactivity (Sinha et al., 2008; Hansen et al., 2014 (part 2 will include a comprehensive modeling study); Zannoni et al., 2015). While studies that employ model OVOC concentrations have a more complete representation of oxidation products (Di Carlo et al., 2004; Edwards et al., 2013), neither of these studies compare measured and modeled OVOC mixing ratios. Additionally, as isoprene hydroxyl hydroperoxide (ISOPOOH) and isoprene hydroxy nitrate (ISPN) standards have only recently become available (Rivera et al., 2014; Lee et al., 2014), first-generation oxidation product measurements are often incomplete.

With high isoprene emissions, the South East United States is an ideal location to reassess questions of missing OH reactivity and speciation of observed reactivity. In addition to measurements of OH reactivity, the 2013 Southern Oxidant and Aerosol Study (SOAS) field campaign provides a comprehensive suite of VOC and OVOC measurements, enabling a more constrained analysis of the contribution from isoprene-derived OVOCs than previously available. This includes first-generation isoprene oxidation products for both low-NO and high-NO oxidation, such as ISOPOOH, ISOPN, isoprene hydroperoxy aldehydes (HPALD), the sum of methyl-vinyl ketone (MVK) and methacrolein (MACR), as well as several



smaller oxidation products. Furthermore, dry deposition rates of isoprene's OVOCs are measured and parameterized for this site (Nguyen et al., 2015), enabling us to reduce some of the uncertainty related to physical losses of carbon. Speciated and total monoterpene measurements provide additional insight into reactive carbon not stemming from isoprene. As a detailed analysis of isoprene photo-oxidation at this site is provided by Su et al. (2015) and Xiong et al. (2015), this work focuses

primarily on the total amount and speciation of OH reactivity above the forest canopy. Using a 0-D box model, we investigate the sources of reactive carbon and compare summation of modeled species with measured OH reactivity. We then discuss our findings in the context of previous studies, and briefly discuss the air quality relevant implications.

# 2 Section (as Heading 1)

## 2.1 SOAS Measurements

Measurements were performed from 1 June to 15 July at the SouthEastern Aerosol Research and CHaracterization (SEARCH) Centreville (CTR) site near Brent, Alabama as part of the 2013 Southern Oxidant and Aerosol Study (SOAS) field campaign (soas2013.rutgers.edu/). CTR is a rural site surrounded by mixed deciduous-evergreen forests, at times experiencing urban-influence from Birmingham, Montgomery, or Tuscaloosa AL. The long term and regional chemical tends observed at this site have been discussed in detail elsewhere (Blanchard et al., 2013; Hidy et al. 2014). We restrict our

analysis to the time frame of good instrumental overlap (11 June to 16 July 2013). All observations shown here are binned to 30 min time intervals. A discussion of missing data interpolation can be found in the supplement.

Table 1 summarizes the chemical measurements used in this analysis and their related uncertainties. Most chemical measurements and solar radiation were acquired from a walk-up tower with a height of ~20 m, approximately 10 m above the forest canopy. CO, Gas Chromatograph-Electron Capture Detector (GC-ECD) measurements and meteorological

parameters (relative humidity, temperature, pressure, and boundary layer height) were acquired from a nearby trailer.

Key measurements to this analysis are OH reactivity, VOCs, and OVOCs. OH reactivity was measured by adding OH to an airstream using a moveable wand, and monitoring the decay of the OH radical by laser-induced fluorescence (Mao et al., 2009). Most VOCs were measured by gas chromatography-mass spectrometry (GC-MS), which provided 5 min samples every 30 min (Gilman et al., 2010). Due to possible line-losses for oxygenated species in GC-MS measurements, Proton-

Transfer-Reaction Time of Flight Mass Spectrometry (PTR-TOFMS, Ionicon Analytik model PTR-TOF 8000) measurements are used for the sum of MVK and MACR (Jordan et al., 2009). The PTR-TOFMS also provided measurements of the total monoterpene mixing ratio. Unspeciated monoterpenes are defined as the difference between PTR-TOFMS measurement of total monoterpenes and the sum of individual species provided by the GC-MS ($\alpha$-pinene, $\beta$-pinene, limonene, mycrene, and camphene).

Glycolaldehyde, ISOPOOH, and isoprene dihydroxy epoxides (IEPOX) were measured by $CF_3O^-$ triple quadrupole chemical ionization mass spectrometry (Paulot et al., 2009; St. Clair et al., 2010; St. Clair et al., 2014). ISOPN, HPALD, the sum of MVK and MACR nitrates ($MACNO_3 + MVKNO_3$), hydroxyacetone, and peroxyacetic acid were measured by chemical





ionization time of flight mass spectrometry (Crounse et al., 2006; Lee et al., 2014). Formaldehyde (HCHO) was measured by fiber-laser-induced-fluorescence, (Hottle et al., 2008; DiGangi et al., 2011), and glyoxal was measured by Laser-Induced Phosphorescence (Huisman et al., 2008). Additional speciated organic nitrates were measured by gas chromatography-electron capture detector (Roberts et al., 2002).

## 2.2 Model simulations

A 0-D box model analysis was performed using the University of Washington Chemical Box Model (UWCM) (Wolfe and Thornton, 2011), incorporating the Master Chemical Mechanism, MVM v3.2 (Jenkin et al., 1997; Saunders et al., 2003, website: http://mcm.leeds.ac.uk/MCM), updated to include the isoprene alkyl radical-$O_2$ adduct equilibria (Peeters and Muller, 2010), isoprene peroxy radical isomerizations (Crounse et al., 2011; da Silva et al., 2010), revised ISOPOOH+OH rate constant (St. Clair et al., 2015), and HPALD photolysis and OH reaction rates (Wolfe et al., 2012). Monoterpenes reactions for species not included in the MCM (i.e., mycrene, camphene, and unspeciated monoterpenes) are described in Wolfe et al. (2011). At each time step, photolysis rates are scaled according to the ratio of measured radiation and the maximum observed radiation at that time of day.

Dry deposition is included for $H_2O_2$, organic hydroperoxides, nitrates, and the isoprene-derived epoxides (IEPOX). Measured deposition velocities are used for $H_2O_2$, IEPOX and ISOPN. For other hydroperoxides and organic nitrates, noontime deposition velocities are calculated according to the relationship with mass shown by Nguyen et al. (2015). Diurnal variability of deposition velocities are scaled according to the measured variation for representative species (ISOPOOH for peroxides, methacrolein nitrate for nitrates).

Dilution is assumed to occur with air with a concentration of zero for all species. This dilution represents entrainment with free tropospheric air and any decrease in concentrations related to unrepresented deposition or advection processes. A constant, empirically determined rate of 4 day$^{-1}$ is used in all analysis presented here, giving a 6 h lifetime with respect to dilution. Sensitivity analysis of this dilution rate is provided in the supplement. The model is initiated with a two day spin up period using diurnal averages of measured species to account for the buildup of unmeasured intermediate species.

Two separate model configurations are used to examine OH reactivity and OVOC concentrations. In all discussions of modeled OH reactivity, OVOC concentrations are constrained to their measurements to ensure the most complete representation of measured OH reaction partners. This includes constraining ISOPN, ISOPOOH, MVK+MACR, $MVKNO_3$+$MACNO_3$, HPALD, and IEPOX by applying modeled isomeric distributions to measured concentrations. Due to partial conversion of ISOPOOH to MVK+MACR in the PTR-TOFMS inlet (Rivera et al., 2014), this represents an upper limit on MVK and MACR measurements. However, because daytime ISOPOOH concentrations are a factor of >5 lower than MVK+MACR, and because the sensitivity to ISOPOOH is only ~30% of that of MVK+MACR, the effect of ISOPOOH on MVK+MACR signal is expected to be negligible. It should be noted that all species that react with OH are included in the calculated reactivity, whereas species that immediately regenerate OH (such as ISOPOOH), would not contribute to measured OH reactivity. However, the contribution from such species is small. The scenario for comparing modeled and





measured OVOCs is identical, except that OVOCs are not constrained. Because OVOC concentrations are calculated in a separate model scenario, any discrepancy between measured and modeled OVOC concentrations does not translate to a discrepancy in calculated reactivity.

In both model configurations, OH, NO, $NO_2$, CO, $O_3$, $H_2O_2$, $HNO_3$, and all primary VOCs are constrained to their measurements. Primary VOCs are defined as any species that are likely to have a significant contribution from direct emissions. This includes alkanes, alkenes, aromatic compounds, and some oxygenated species (methanol, ethanol, acetone, methyl-ethyl-ketone, acetaldehyde, biacetyl, propanal, hydroxyacetone, and formic acid). Table 1 provides a listing of constraints for each model scenario.

## 3 Results

### 3.1 Measured and Modeled OH reactivity

Figure 1 shows a comparison between measured and modeled OH reactivity for the constrained-OVOC scenario. Model and measured values are well correlated ($r^2$=0.85), with a slope of 0.80 ± 0.02. The average missing reactivity for all measurement points is 16 ± 18%. An uncertainty of 20% is assigned to model reactivity based on the uncertainty in isoprene, which comprises the majority of modeled reactivity. Propagating measurement uncertainty (20%) and model uncertainty (20%) yields at least 28% uncertainty in the missing fraction of OH reactivity. As both the slope and average discrepancy agree with measurement within 28%, on average, we find no significant discrepancy between modeled and measured OH reactivity. A subset of points that correspond to high β-pinene concentrations fall outside of this range. Most of these points occur early in the measurement period, from 11 June – 17 June. To investigate the sources of these discrepancies, we examine both the diurnal variability and composition of OH reactivity.

Figure 2 shows the diurnal variability of the missing portion of reactivity. In the afternoon, the model typically captures >90% of OH reactivity. At night, the model typically captures ~80% of measured reactivity. Early morning discrepancies show the largest average discrepancies, reaching an average of 32% missing reactivity at 7:00 L.T.

The average diurnal speciation of observed reactivity is shown in Figure 3. Primary biogenic VOCs make up the largest fraction of modeled OH reactivity throughout the entire day, with isoprene contributing ~60% in the afternoon and ~30-40% at night, and monoterpenes contributing ~15-25% at night. Oxygen containing VOCs contribute less significantly at all time points (~20-28%), and the largest individual contributors are measured species such as HCHO (~3-4%), MVK, and MACR (~2-4%). Unmeasured oxidation products contribute ~6-10% of total modeled reactivity, and are most prominent at night.

As discussed in Edwards et al. (2013), the increase in total reactivity with increase in isoprene is another useful parameter when considering OH reactivity speciation. In a plot of total OH reactivity plotted against the contribution from isoprene alone, the slope is related to the contribution from short-lived isoprene-derived OVOCs and VOCs co-emitted with isoprene. Figure 4 shows this relationship for measured and modeled OH reactivity, still referring to the OVOC-constrained scenario. Both observed and modeled OH reactivity are tightly correlated with OH reactivity from isoprene ($r^2$≥0.81). The slopes are





not much larger than one, again demonstrating that isoprene (rather than its oxidation products) dominates daytime reactivity at this site. The difference between model ($1.22 \pm 0.02$) and observed ($1.43 \pm 0.02$) slope is small but significant. This amounts to 15% of reactivity correlated with isoprene reactivity not captured by measured species or modeled unmeasured oxidation products. The y-intercept from measurements ($6.3 \pm 0.1$ s$^{-1}$) and model ($5.2 \pm 0.1$) also show a small but significant

difference. This indicates a missing reactivity of ~1 s$^{-1}$ that is temporally distinct from isoprene reactivity.

## 3.2 Measured and Modeled OVOCs

By investigating the model's ability to capture measured OVOC concentrations, we can determine the reliability of model predictions of unmeasured species. As the model is constrained to measured OVOC concentrations when calculating model OH reactivity, the contribution of unmeasured species to total reactivity will be different in these two scenarios. However,

the assessment of model performance can be extended to the constrained-OVOC scenario.

Figure 5 shows the model's prediction of several measured OVOC concentrations. Isoprene's first generation oxidation products MVK+MACR, ISOPOOH, ISOPN, and HPALD are over-predicted in the afternoon. Though the uncertainties in each of these measurements is large (40-70%), all model concentrations are much higher than measurements. The model overestimates daytime HPALD observations by a factor of ~ 6, ISOPOOH by a factor of ~4, and ISOPN by a factor of ~ 3.

This translates to an overprediction of IEPOX and MACNO$_3$+MVKNO$_3$, which are formed in the oxidation of ISOPOOH and ISOPN, respectively. For MVK+MACR, the daytime over-prediction is approximately a factor of two. Daytime agreement for MPAN, which is formed from MACR, is comparatively good. In general, smaller oxidation products (i.e., glyoxal, glycolaldehyde, and HCHO) are less susceptible to overprediction.

Model OVOC concentrations are highly sensitive to the assumed dilution scheme (see full discussion in Supplement).

However, in an investigation of isoprene photochemistry and turbulent mixing during this campaign, a more complex mixed layer chemical model (MXLCH) predicts similar peak values for ISOPOOH (1.5 ppb), MVK+MACR (3.0 ppb), and ISOPN (80 ppt) in the convective mixed boundary layer (Su et al., 2015). The MXLCH MVK+MACR mixing ratios are substantially higher than ground-based measurements, but comparable to measurements from the Long-EZ research plane flying at altitudes from 100 – 1000 m a.g.l. Flight based measurements were relatively uniform throughout the boundary

layer. In order for dilution alone to account for the low concentrations of first generation oxidation products, extremely high dilution rates would need to be incorporated. For ISOPOOH, a constant rate of 40 day$^{-1}$, (roughly five times the photochemical loss rate) would be needed. Most importantly, when OVOCs are constrained, the assumed dilution scheme has very little effect on the model OH reactivity (Figure S2), as measured species dominate total reactivity.

## 4. Discussion

While on average the model largely captures the absolute value of OH reactivity at SOAS (Fig 1), there are small but significant differences (15%) in the increase of total reactivity and reactivity from isoprene alone (slope of Fig 3). While





most measured species have uncertainties >15%, it is unlikely that all measured species are systematically low, suggesting this discrepancy is likely the result of unmeasured species. When given a constrained precursor, the model either reproduces or overpredicts the resulting oxidation products (Fig 5, Fig S4). As isoprene and its oxidation products are heavily constrained, we conclude the unmeasured primary species co-emitted with isoprene (and those species' oxidation products) are the likely source of this small discrepancy.

As observed daytime isoprene concentrations increase with temperature, the difference in slope also represents a temperature-dependent daytime missing reactivity. The temperature dependence observed at SOAS is greater than that observed by Di Carlo et al. (2004) and the dependence of monoterpene emissions (Figure 6). Emissions which depend both on temperature and light are likely to have stronger net temperature dependence, as temperature increases with increasing solar radiation. Therefore, a portion of the total missing emissions is likely characterized by both a light and temperature dependence.

Furthermore, the model is missing ~1 s$^{-1}$ reactivity that is temporally unrelated to the oxidation of isoprene and co-emitted species (Figure 3). This is consistent with the diurnal variability of missing reactivity, with larger portions occurring at night and in the early morning (Figure 2). Likely, missing nighttime reactivity is composed of a mixture of unmeasured primary emissions, unmeasured oxidation products, and long-lived unmeasured species mixed in from the residual layer. Xiong et al. (2015) show that 27% of the early morning increase in ISOPN results from downward mixing from the residual layer during this campaign. Similarly, there may be unmeasured OH reaction partners stored in the nocturnal boundary layer that lead to an increase in OH reactivity upon breakup of the inversion. Like β-pinene, anthropogenic VOCs such as toluene and benzene are highest at night. However, these species were not unusually high during the 11 June – 16 June period which demonstrated the highest missing reactivity, and therefore unmeasured anthropogenic VOCs are unlikely the major source of discrepancy. Sesquiterpenes ($C_{15}H_{24}$) are another class of VOC which typically follow the emission patterns of monoterpenes. The total sesquiterpene emission rate from broadleaf trees is estimated to be ~67% the emission rate of total monoterpenes in terms of total mass (Sakulyanontittaya et al., 2008). Assuming a reaction rate with OH of β-caryophyllene, ~200 ppt of sesquiterpenes would provide the 1 s$^{-1}$ offset in reactivity temporally separated from isoprene.

Much like the previous work of Zannoni et al. (2015), we find good daytime agreement between measured and modeled reactivity above the forest canopy, and that the majority of reactivity can be attributed to primary emissions. Using measurements of first and later generation OVOCs as a constraint on the amount of total unmeasured oxidation products, we find no evidence of substantial contributions of unmeasured OVOCs to above-canopy OH reactivity. This is in contrast to studies of Edwards et al. (2013) and Hansen et al. (2014), who showed that these species may contribute significantly to OH reactivity directly above the forest canopy. Varying amounts of intra-canopy oxidation are likely to result in these different conclusions, as secondary compounds will quickly become more important than the primary isoprene emissions at higher altitudes or farther downwind of the forest.

Based on measured OH concentrations, the measured concentrations of OVOCs suggest surprisingly little intracanopy oxidation of primary VOCs at this site. Furthermore, advection does not appear to bring in processed isoprene emissions.



Despite measuring ~10 m above the forest canopy in a relatively homogeneous area, OH reactivity is primarily composed of measured primary species. Our model overpredicts concentrations of isoprene's first generation oxidation products by at least a factor of two. If these species and other OVOCs were not constrained by measurements, these overpredictions would lead to problematic conclusions about the speciation of reactivity. In the relationship of reactivity from isoprene to total

reactivity, the modeled slope (1.43 ± 02) and measured slope (1.44 ± 0.02) would show no discrepancy. While the true observed missing contribution is small, it highlights the contribution from primary species whose oxidation maybe important downwind.

## 5. Conclusions

In summary, the discrepancies in the absolute value of measured and modeled OH reactivity are rarely significant at this site. This suggests that the total $RO_2$ production rate and resulting $O_3$ formation are likely well understood. In contrast, small but significant discrepancies in the observed and calculated trend in OH reactivity with increasing isoprene suggest missing sources of reactive carbon. The model fails to capture a portion of reactivity that is temporally related to isoprene, as well as a portion unrelated to local isoprene oxidation. As isoprene oxidation products are heavily constrained and the model does

not typically underestimate OVOCs, we propose that missing primary emissions and their oxidation products are likely candidates for both sources of reactive carbon. While these missing emissions do not lead to significant inconsistencies between measured and modeled OH reactivity, at larger total emissions, the trending discrepancy may lead to larger missing fractions of OH reactivity.

Additionally, the speciation of this missing carbon source has air quality relevant implications. For example, though

monoterpenes are much less abundant than isoprene, they can substantially effect SOA formation. Ayres et al. (2015) found that organic nitrate aerosol from $NO_3$ + monoterpenes is a substantial contribution to observed particulate matter at this site, with a SOA molar yield of 23-44%. In contrast, the comparable isoprene nitrate is primarily a gas-phase product. Through positive matrix factorization analysis of aerosol mass spectrometer measurements, Xu et al. (2015) found monoterpene + $NO_3$ chemistry contributes 50% to total nighttime organic aerosol formation at this site, whereas IEPOX-derived SOA

constitutes 19-34% total organic aerosol. Additionally, Su et al. (2015) cite aerosol uptake and condensed phase reactivity as a possible explanation for the large discrepancy between observed and modeled ISOPOOH at this site, which implies a large loss of total carbon to the aerosol phase. While the magnitude of OH reactivity is well captured, continued efforts in speciated OVOC and VOC measurements are vital to fully understand the SOA contribution from various primary emissions.

## Acknowledgements

The authors would like to acknowledge contribution from all members of the SOAS science team. Funding was provided by U.S. EPA-Science to Achieve Results (STAR) program- Grant 83540601. A.H. Goldstein and P.K. Misztal acknowledge



support from EPA STAR Grant R835407. This research has not been subjected to any EPA review and therefore does not necessarily reflect the views of the Agency, and no official endorsement should be inferred. Additional funding was provided by NSF-grant AGS-1247421. J. Kaiser acknowledges support from NASA Headquarters under the NASA Earth and Space Science Fellowship Program - Grant NNX14AK97H.

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



Table 1. SOAS measurements used in this study

| Instrument | Parameters[a] | 1 σ uncertainty | Reference/Model number |
|---|---|---|---|
| OH Laser-induced fluorescence | OH[b] <br> OH reactivity | 16% (30 min) <br> 10% (30 sec) | Mao et al., 2009 |
| Tropospheric Airborne Chromatograph for Oxy-hydrocarbons | VOCs[b] | 20% (30 min) | Gilman et al., 2010 |
| Proton-transfer-reaction time-of flight mass spectrometer | Total monoterpenes[b] <br> MVK+MACR | 20% (1 min) <br> 40% (1 min) | Jordan et al., 2009 |
| $CF_3O^-$ triple quadrupole chemical ionization mass spectrometry | ISOPOOH, IEPOX, Glycolaldehyde | 100 ppt + 70% (0.5 sec) | St. Clair et al., 2010 |
| Fiber-Laser-Induced-Fluorescence | HCHO | 15% (1 s) | Hottle et al., 2008; DiGangi et al., 2011 |
| Madison Laser-Induced-Phosphorescence | Glyoxal | 9%(1 s) | Huisman et al., 2008 |
| Gas Chromatograph-Electron Capture Detector | PAN, PPN, MPAN | 20% (20 min) | Roberts et al., 2002 |
| $CF_3O^-$ compact time of flight mass spectrometer | HCOOH[b], $H_2O_2$[b], $HNO_3$[b], ISOPN, Hydroxyacetone, Peroxyacetic acid, HPALD, $MACNO_3$+$MVKNO_3$ | 100 ppt + 30-50% (5 sec) | Crounse et al., 2006 |
| Absorption of IR with Gas Filter Correlation | CO[b] | 7.4% (5 min) | Thermo Scientific Model 48i-TLE |
| Nitrogen Oxides by Cavity Ring Down | $O_3$[b] <br> NO[b] <br> $NO_2$[b] | 3% (1 min) <br> 8% (1 min) <br> 3% (1 min) | Fuchs et al., 2009; Wild et al., 2014 |

[a]All species listed are constrained when calculating OH reactivity.

[b]Denotes species constrained when calculating both OH reactivity and OVOC mixing ratios.



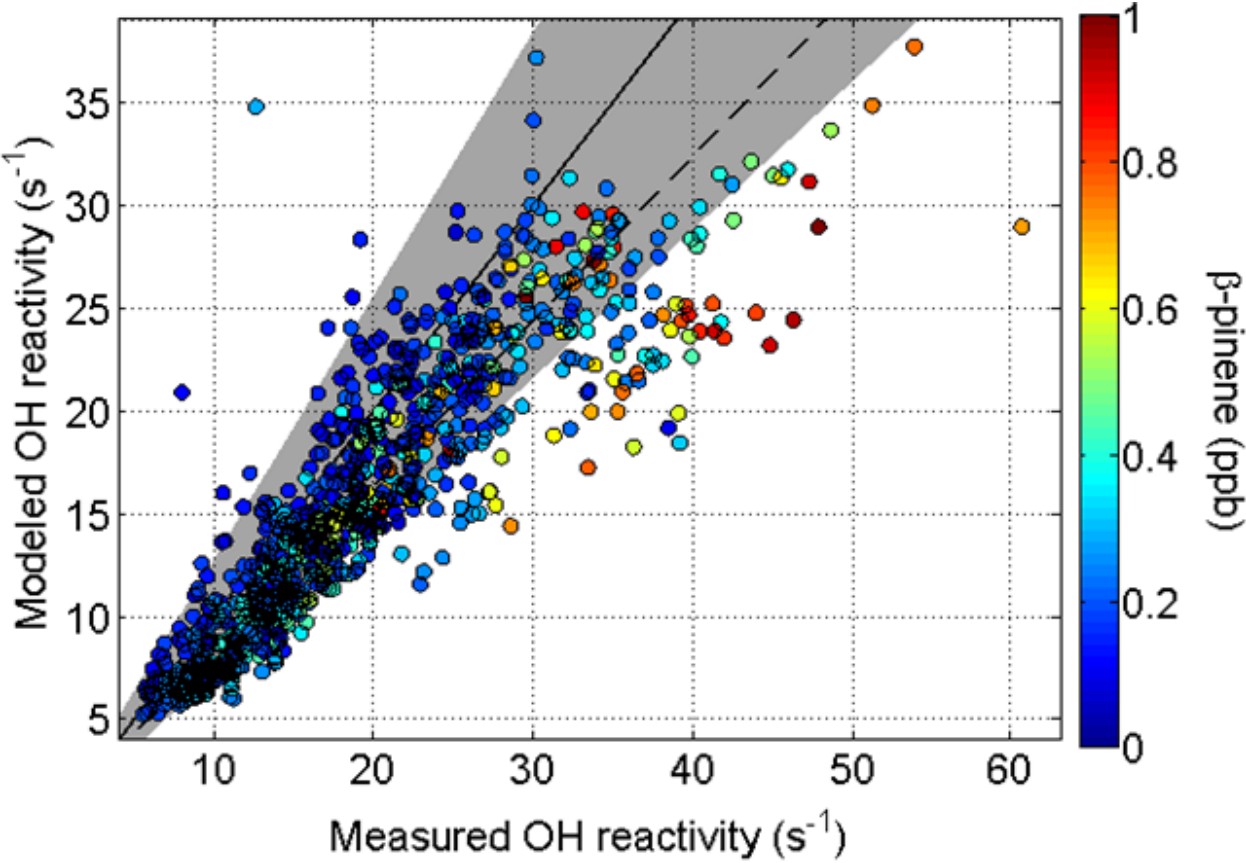

Figure 1. Comparison of measured and modeled OH reactivity colored by measured β-pinene concentrations. The solid line represents 1:1 agreement, and the dashed line represents the linear least squares fit weighted by uncertainty (York et al., 2004; Thirumalai et al., 2011). The grey shaded area represents points within combined uncertainty of 1:1 agreement (± 28%).





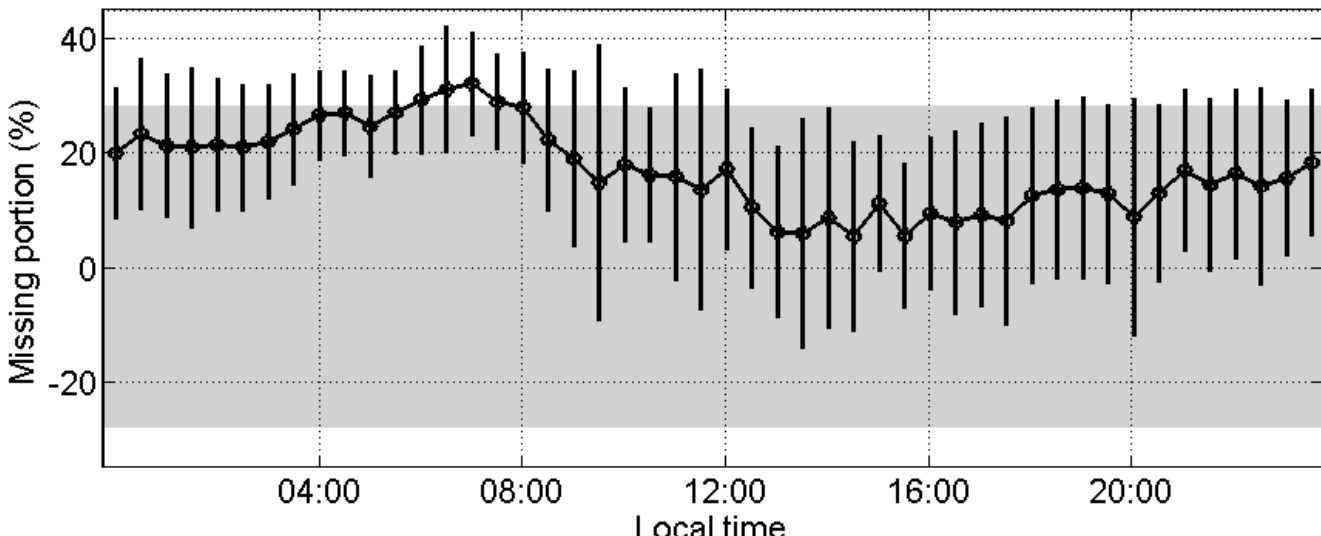

Figure 2. Diurnal profile of the discrepancy between measured and modeled OH reactivity. Error bars represent 1 σ standard deviation of diurnal variability. Points the in gray shaded area within the range of agreement considering combined measurement and model uncertainty (±28%).





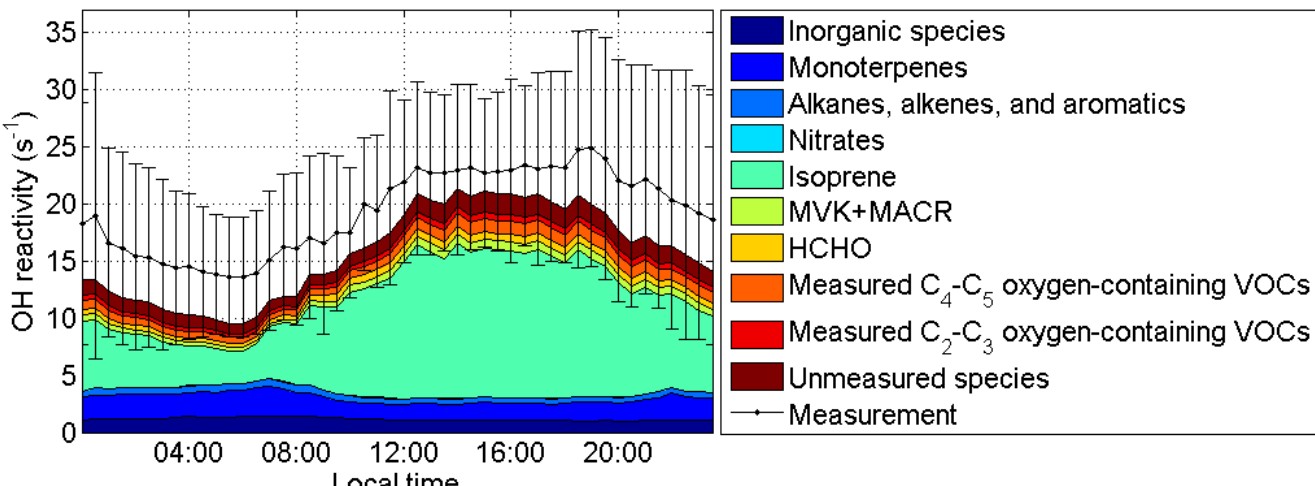

Figure 3. Diurnal profile of measured and modeled OH reactivity. Error bars represent 1 σ standard deviation of diurnal variability.





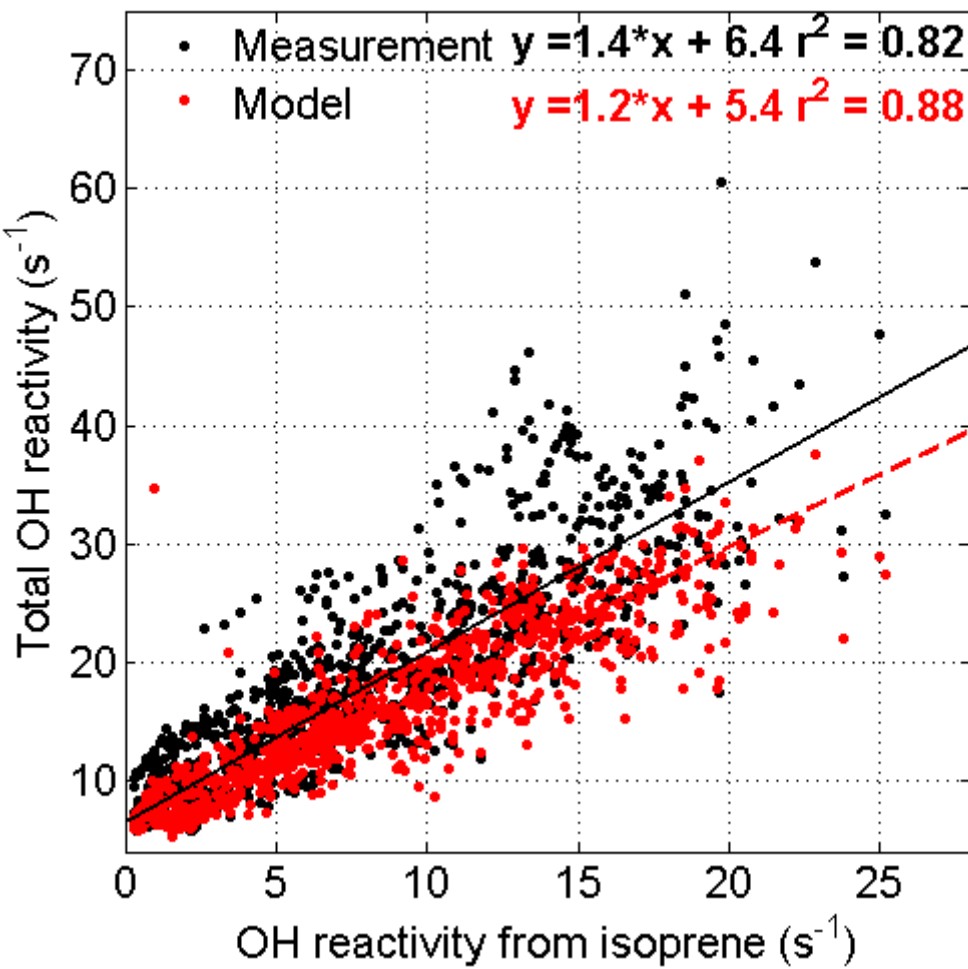

Figure 4. Total measured and modeled OH reactivity as a function of the OH reactivity calculated from isoprene alone. Lines represent least square linear fits weighted by uncertainty for measured (solid) and model (red dashed) OH reactivity.



Figure 5. Average diurnal profile mixing ratios for isoprene, β-pinene, and several measured oxidation products calculated by the model. Error bars and shaded area represent 1 σ standard deviation of diurnal variability. For each species, model results are not included for points where measurements are missing.





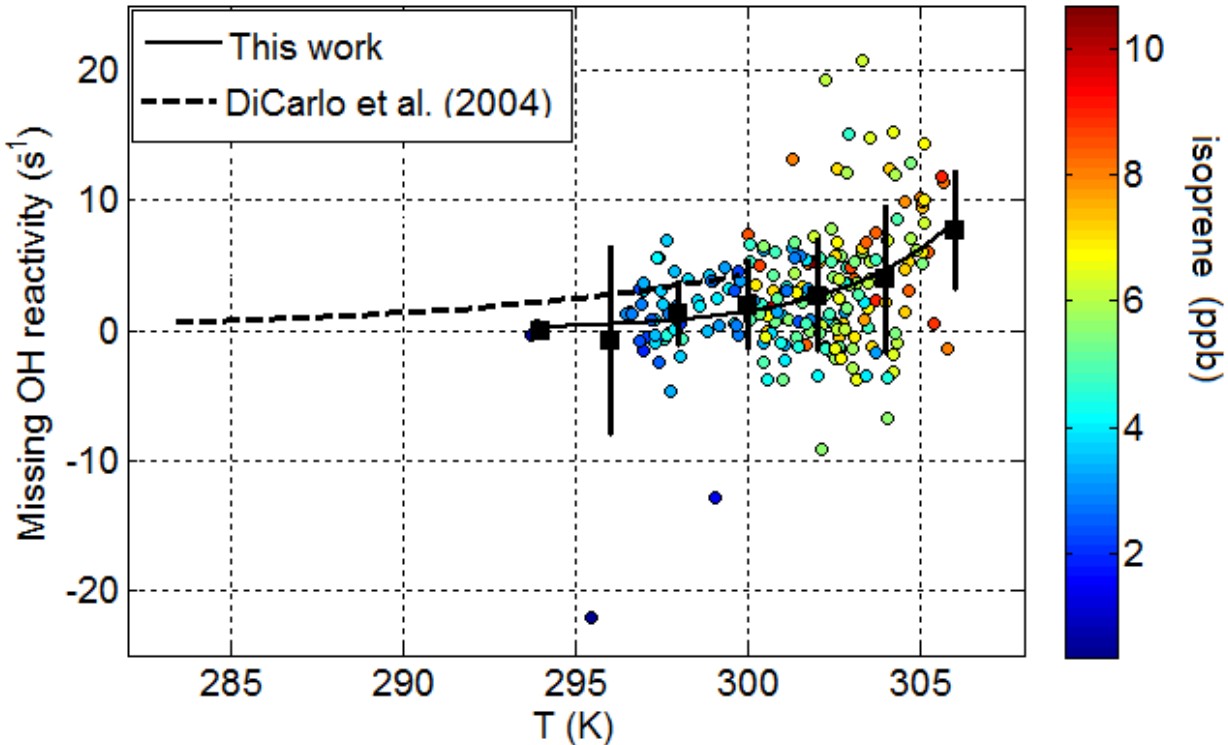

Figure 6. Daytime (10:00-16:00 L.T.) missing reactivity as a function of temperature and isoprene. Black squares represent 2 degree averages and standard deviations. All daytime points are fit according to the function $y=\alpha*\exp(\beta(x-293))$. The temperature dependence observed at SOAS ($\beta = 0.30$) is greater than that observed by Di Carlo et al. (2004) and the dependence of monoterpene emissions ($\beta = 0.11$).