# Peer review of "Speciation of OH reactivity above the canopy of an isoprenedominated forest"

_Atmospheric Chemistry and Physics, 2015_

## Referee Comment (RC1) · Anonymous Referee #1 · 21 Feb 2016

Measurements of OH reactivity are made during the SOAS study in SE USA, at an isoprene dominated site, and compared with a model. The model is constrained with measured OH radicals, and also an impressive range of OH sinks, including those VOCs directly emitted by plants, and importantly for this environment, oxidation products, particularly oxidation products of isoprene, which is the dominant emitted species in this environment. The model uses a very detailed chemical mechanism (MCM v3.2) also includes updates in the chemistry from RO2 equilibration and RO2 H-atom shift isomerisations, as well as updated kinetics and photolysis rates of isoprene oxidation products. There is generally good agreement between the model and measured OH reactivity (average slope of 0.8 model to measured, 16 (+/- 20%) missing OH reactivity). So when measurement uncertainty is also factored in (20%) the model-measurement difference is not significant and it is not necessary to invoke that there is a significant

amount of missing OH reactivity, and hence associated uncertainties in identifying what the missing OH reactivity is. These uncertainties can have important implications for calculating overall rates of VOC oxidation, and hence the production of ozone, SOA and other policy-related parameters. The difference between modelled and measured OH reactivity during an average diurnal profile is also discussed, with largest discrepancies observed in the early morning (30%). In fact the largest contribution towards the OH reactivity is calculated to be the primary biogenic emissions rather than oxygenated products.

An important conclusion from this study is that if these oxygenated products had not been measured, the model would have overpredicted the OH reactivity, leading to difficult conclusions, as the model used in this study significantly overpredicts the concentration of OVOCs (by large factors). The good agreement is achieved by being able to constrain the model with measured concentrations of isoprene oxidation products, which have been absent from many other studies, and this is a real strength of this paper. Measurements of ISOPOOH, ISOPN and isoprene HPALDS are novel and greatly aid the interpretation of OH reactivity in this environment. The fact that these OVOCs are actually measured is shown to be crucial, not because they are a major measured OH sink at this site, but because the model used in this study is not able to calculate the concentration of these oxygenated products with any degree of accuracy, e.g. the model overestimates by a factor of 6 for HPALD, and a factor of 4 for ISOPOOH. If the model had used the modelled OVOCs to calculate OH reactivity (rather than constrained to them), the overprediction of OH reactivity would have led to different conclusions. Reasons for the model overprediction of the OVOCs may be uncertainties in dilution rates following emission, which are discussed in the supplement. As the model overpredicts the OVOCs, the model underprediction of OH reactivity (when model is constrained to measured OVOCs) suggests that this is due to unmeasured primary emissions, and not missing OVOCs. A plot of measured OH reactivity versus that specifically for reaction with isoprene itself shows a slope of 1.43 showing that isoprene rather than its oxidation products dominates OH reactivity at this site.

Significant here is that the paper notes that the rates of deposition of these OVOC species to the canopy, which is often a model parameter for which there is considerable uncertainty, have been measured by other groups in the SOAS study, and therefore does not have to be estimated in the model. A sensitivity study is presented on the impact of changes in the dilution factor on the OH reactivity.

The paper has some very interesting findings and the measurements are of good quality. The conclusions of the paper are significant for this type of environment, and are different to some earlier campaigns in a similar type of environment. However, some statements do not appear consistent with the data presented, and some further detail is missing in places. The authors should respond to the following points.

Abstract

(1) The abstract does not contain any quantitative information about the level of agreement between model and measured OH reactivity, which is the main result. There is a long introduction to the abstract and it is not until line 18 that any points specifically relevant to the results of this study are presented. Lines 10-17 need to be moved to the introductory material in the main paper, and some of the main results from the study (overall level of agreement (i.e. that the model underpredicts), slopes of main correlation plots, OH reactivity versus isoprene calculated reactivity, diurnal behaviour, statement that there are large model overprediction of OVOCs with values etc.) need to be stated. The abstract needs to be extended considerably in terms of summarising the main results.

Paper

(1) Page 7 (there are no page numbers, so pages here refer to pages with the cover page as page 1) lines 2-3. In the measurement of OH reactivity, how was the zero of the instrument determined, and what is the value? Also, presumably in this environment the level of NO is low enough that any corrections for recycling of OH from $HO_2+NO$ within the sampling airstream are not necessary? If this is the case it should be stated.

It would also be worth stating how the accuracy of the instrument is checked using known OH sinks.

(2) Page 7, line 25, MCM (not MVM).

(3) Page 8, line 29. Would any process immediately regenerate OH? The timescale may be fast compared with the OH decay, but the actual values should be compared, and one shown to be much faster than the other, rather than just this statement.

(4) Page 9, line 1. Small is subjective, please give a % value here to evidence this.

(5) Although not the subject of this paper (and measured OH is used to constrain the model), it would be useful to state the level of agreement between the model and measured OH and other measured radicals (HO2 and maybe RO2). It would be useful just to confirm how the model performs for these species (given that the model comparison for OVOCs is discussed later).

(6) Page 10, line 1. The largest discrepancy is observed at 0700 LT (32%). Can the authors comment on the general shape of Figure 2?

(7) Page 10, line 16. It is stated that the slopes are not much larger than one. This is rather subjective as the observed slope of measured OH reactivity versus that from isoprene itself is 1.43, which although not a factor of 3, 5 etc., is significantly greater than 1 (by almost 50%). This statement ought to be qualified. It can be seen clearly from Fig 3 that at times the reactivity due to isoprene is considerably less than the total measured OH reactivity.

(8) Page 12, line 3. Not slope of fig 3? You mean slope of fig 4?

(9) It is difficult to compare the temperature dependence observed with that of Di Carlo et al., as the range of values of temperature only seem to have a limited range where they overlap. The parameterisation of Di Carlo seems to stop at 300 K, and in this region the current work's function does not change much. It is only above 300 K that the function for this work becomes significantly steeper than that of Di Carlo. Some further

discussion of this is needed, and the parameterisation of Di Carlo needs extending to higher temperatures compare easily.

(10) Page 12. The paper states that "…the model is missing $\sim$ 1 s-1 that is temporally unrelated to the oxidation of . ….. (Figure 3)". However, inspection of Figure 3 suggests that the difference between the measured OH reactivity (black points) and the cumulative modelled reactivity is more than 1 s-1? It looks more like between 2 and 5 s-1 (depending on time of day). Some modification of this statement is therefore needed.

(11) Page 12, line 12, missing "of"

(12) Page 13, line 2. Again 1s-1 seems to be a significant underestimate of the difference between measured and modelled shown in Figure 3. The amount of sesquiterpenes would therefore need to be more than $\sim$ 200 ppt.

(13) Page 13, line 22. The values quoted here are not consistent with the values quoted on page 10. The measured slope was quoted as 1.43 there, and modelled slope 1.22. Also, it says +/- 02 (should be 0.02).

(14) Table 1. Glyoxal row, space between 9% (1 s)

(15) Figure 1. Please plot this graph from the origin (0,0) as it will be more informative. At present it is rather misleading as it suggests the points go to the origin.

(16) Figure 1. The slope of the linear least squares fit weighted by uncertainty must be stated in the caption. This is one of the most important results of the paper.

(17) Caption for Fig 2, second line, "Points the in gray…" needs correction

(18) Figure 3. As commented above the difference is considerably more than 1s-1 quoted in the text (seems to be 2-5 s-1 depending on time of day). Suggest reversing the order of the legend. At present the measurement is at the bottom, with Inorganics at the top, whereas the figure is measurement at the top, and Inorganic at the bottom.

(19) Figure 4. Given the text gives the slopes at 1.43 and 1.22 (note inconsistency with

page 13), the equation given on the figure needs to reflect this quoted accuracy. What does "weighted by uncertainty" mean?

(20) Figure 6. See comments above about the degree of overlap in T for the solid and dashed lines. What is the value of Greek alpha for each?

---

## Referee Comment (RC2) · Anonymous Referee #2 · 28 Feb 2016

This paper presents measurements of total OH reactivity with measurements of many OH radical sinks, including several isoprene oxidation products during the SOAS campaign. In contrast to many previous studies, the authors find that the modeled OH reactivity agrees well with the measured total OH reactivity when the model is constrained to the measured OH, VOCs, and OVOCs. However, when the OVOCs are unconstrained, the model tends to overestimate the mixing ratios of OVOCs, including isoprene oxidation products MVK+MACR, ISOPOOH, IEPOX and HPALD. Because the model tends to overpredict OVOCs, the authors conclude that the observed missing reactivity in the morning at this site is not due to unmeasured oxidation products, but due to unmeasured primary emissions and their oxidation products.

The paper demonstrates the importance in measuring these isoprene oxidation products in order to constrain models and understand OH reactivity. It also appears to

demonstrate the inability of current models to accurately model these OVOCs. Unfortunately, the paper treats the modeling of the OH reactivity and the modeling of the OVOCs separately. However, it appears that if the OVOCs were not measured, the model would overestimate the observed OH reactivity, bringing into question the ability of models to fully reproduce the observed OH reactivity and therefore $RO_2$ and $O_3$ production.

Overall, this is an interesting paper that does provide some new information regarding the nature of missing OH reactivity in forest environments. The main thrust of the paper is that the modeled reactivity agrees well with the measured reactivity when constrained to measured OVOCs and that primary emissions dominate the OH reactivity at this site, in contrast to previous studies in similar environments. However this conclusion is not highlighted clearly enough in the paper and the authors attempt to extend this conclusion to our understanding of $RO_2$ production and $O_3$ production without providing sufficient modeling studies to support this conclusion. The fact that the model significantly overestimates the measured OVOCs suggests that our understanding of $RO_2$ chemistry and $O_3$ production is still incomplete. Unfortunately, there is little discussion as to potential reasons why the model significantly overestimates the observed OVOCs, although the paper mentions that uncertainties associated with modeled dilution rates of these species may be responsible. The paper would benefit from an expanded discussion of potential reasons for the model overprediction of OVOCs and their implications.

Specific comments:

1) As discussed above, a main conclusion of the paper appears to be that because the modeled OH reactivity (constrained to the OVOCs) agrees well with the measured OH reactivity that the total $RO_2$ production rate and therefore $O_3$ production is well understood (page 9 lines 10-11). However, the fact that the model overestimates isoprene oxidation products and other OVOCs suggests that when the model is not constrained to measurements of these compounds that the model may not be able to reproduce

total $RO_2$ production, as many of these OVOCs are produced from $RO_2/HO_2$ chemistry, such as ISOPOOH and HPALD. The authors need to provide more information to justify this conclusion.

2) One test of the ability of the model to reproduce the observed OH reactivity would be to unconstrain the model to the measured OH and the measured OVOCs. It's not clear from the information given in the paper whether unconstraining the model to the measured OH impacts the modeled OH reactivity as the paper does not state how well the model is able to reproduce the measured OH when constrained to the measured species, or how sensitive the modeled OH reactivity is to the OH concentration. The paper would be stronger if the authors provided several model scenarios to compare to the measured OH reactivity, such as i) a base scenario where the model is constrained only by the traditional measured VOCs, NOx, etc., ii) a scenario where the model is further constrained by the measured OH but not constrained by the measured OVOCs, and iii) the scenario where the model is constrained by all measured species.

3) It is surprising that the authors chose to constrain the model to the measured OH but not to the measured $HO_2$, given the importance of peroxy radical chemistry to the formation of OVOCs such as ISOPOOH and HPALD. Were $HO_2$ measurements not available? Does constraining the model to the measured OH reproduce the measured $HO_2$? Or does this model scenario overestimate $HO_2$ leading to the overestimation of the observed OVOCs? If the model is constrained to both measured OH and $HO_2$, are the modeled OVOCs in better agreement with the measurements? These and other tests of the model would provide important information regarding the reasons for the model's inability to reproduce the observed OVOCs.

---

## Author Comment (AC1) · 18 Apr 2016

**Response to interactive comments on "Speciation of OH reactivity above the canopy of an isoprene-dominated forest" by J. Kaiser et al.**

We thank the reviewers for the thorough reading and helpful comments. Below, we address each remark individually. Original comments are in green, and our response follows, indented and in black. Our pagination refers to the pdf of the discussion paper. Beyond the changes to the manuscript outlined below, please note a change in supplement "entrainment" scenario which corrects a previous error in the calculations for the dilution sensitivity test. These changes have no impact on our analysis or conclusions.

**Referee #1**

(1) The abstract does not contain any quantitative information about the level of agreement between model and measured OH reactivity, which is the main result. There is a long introduction to the abstract and it is not until line 18 that any points specifically relevant to the results of this study are presented. Lines 10-17 need to be moved to the introductory material in the main paper, and some of the main results from the study (overall level of agreement (i.e. that the model underpredicts), slopes of main correlation plots, OH reactivity versus isoprene calculated reactivity, diurnal behaviour, statement that there are large model overprediction of OVOCs with values etc.) need to be stated. The abstract needs to be extended considerably in terms of summarising the main results.

We have substantially changed the abstract to eliminate much of the background and add a more quantitative and detailed summarization. The new abstract reads:

Measurements of OH reactivity, the inverse lifetime of the OH-radical, can provide a topdown estimate of the total amount of reactive carbon in an airmass. Using a comprehensive measurement suite, we examine the measured and modeled OH reactivity above an isoprene-dominated forest in the South East United States during the 2013 Southern Oxidant and Aerosol Study (SOAS) field campaign. Measured and modeled species account for the vast majority of average daytime reactivity (80-95%), and a smaller portion of night-time and early morning reactivity (68-80%). The largest contribution to total reactivity consistently comes from primary biogenic emissions, with isoprene contributing ~60% in the afternoon, ~30-40% at night, and monoterpenes contributing ~15-25% at night. By comparing total reactivity to the reactivity stemming from isoprene alone, we find that ~20% of the discrepancy is temporally related to isoprene reactivity, and an additional constant ~1 s-1 offset accounts for the remaining portion. The model typically overestimates measured OVOC concentrations, indicating that unmeasured oxidation products are unlikely to influence measured OH reactivity. Instead, we suggest that unmeasured primary emissions may influence the OH reactivity at this site. While the magnitude of OH reactivity is related to RO2 production and subsequent ozone formation, determining the molecular structure of compounds related to missing reactivity is essential to understanding its impact.

Paper

(1) Page 7 (there are no page numbers, so pages here refer to pages with the cover page as page 1) lines 2-3. In the measurement of OH reactivity, how was the zero of the instrument determined, and what is the value? Also, presumably in this environment the level of NO is low enough that any corrections for recycling of OH from HO2+NO within the sampling airstream are not necessary? If this is the case it should be stated. It would also be worth stating how the accuracy of the instrument is checked using known OH sinks.

The operating procedures for the OH reactivity instrument are described in Mao et al. (2009). The OH reactivity instrument zero is determined by measuring the wall loss of the OH radical while using a clean carrier gas. The uncertainty in the zero is  $0.5 \text{ s}^{-1}$ , with 2  $\sigma$  confidence. The recycling of OH from HO2+NO was corrected by taking into account measured HO2 decays. The accuracy of the instrument was verified using gasses with well-known reaction rate coefficients (C3F6 in the field, and CO, propane, propene, and isoprene in the lab).

In the manuscript, we now state the value of the instrument offset during SOAS and refer the reader to Mao et al. (2009) for a description of the measurement technique.

**(2) Page 7, line 25, MCM (not MVM).**

This has been corrected.

(3) Page 8, line 29. Would any process immediately regenerate OH? The timescale may be fast compared with the OH decay, but the actual values should be compared, and one shown to be much faster than the other, rather than just this statement.

According to the version of the MCM used here, some processes immediately regenerate OH, most notably the reactions of some hydroperoxides with OH. For example, the product of ISOPOOH + OH reaction is IEPOX + OH. Because loss and production of OH is simultaneous, the ISOPOOH + OH reaction would not impact the measured OH decay. We have clarified this in the manuscript.

(4) Page 9, line 1. Small is subjective, please give a % value here to evidence this.

We now state that the average total calculated reactivity of species that immediately regenerate OH is  $0.6 \pm 0.3 \text{ s}^{-1}$ .

(5) Although not the subject of this paper (and measured OH is used to constrain the model), it would be useful to state the level of agreement between the model and measured OH and other measured radicals (HO2 and maybe RO2). It would be useful just to confirm how the model performs for these species (given that the model comparison for OVOCs is discussed later).

We cannot provide a comparison of measured and modeled OH or  $RO_2$  because our model is constrained to measured OH values and  $RO_2$  measurements are not available.

 $HO_2$  was not constrained and can be compared to measurements. Our results are shown below. A manuscript comparing measured and modeled OH and  $HO_2$  has recently been submitted to J. Atmos. Sci. (Feiner et al. 2016). Slightly different model configurations and selection of days results in a small difference between studies, but both show similar results for  $HO_2$ . Feiner et al. also find good agreement for OH (chemically zeroed measurement) and model values. We direct readers to the Feiner et al. paper for a more complete discussion of the  $HO_X$  budget for this campaign.

(6) Page 10, line 1. The largest discrepancy is observed at 0700 LT (32%). Can the authors comment on the general shape of Figure 2?

The general shape of Figure 2, including the peak of missing reactivity in the early morning, is addressed in the discussions section (page 8, starting at line 13).

(7) Page 10, line 16. It is stated that the slopes are not much larger than one. This is rather subjective as the observed slope of measured OH reactivity versus that from isoprene itself is 1.43, which although not a factor of 3, 5 etc., is significantly greater than 1 (by almost 50%). This statement ought to be qualified. It can be seen clearly from Fig 3 that at times the reactivity due to isoprene is considerably less than the total measured OH reactivity.

This sentence has been eliminated, as the numbers themselves adequately convey the relationship between total reactivity and isoprene.

(8) Page 12, line 3. Not slope of fig 3? You mean slope of fig 4?

This has been corrected to refer to Figure 4.

(9) It is difficult to compare the temperature dependence observed with that of Di Carlo et al., as the range of values of temperature only seem to have a limited range where they overlap. The parameterisation of Di Carlo seems to stop at 300 K, and in this region the current work's function does not change much. It is only above 300 K that the function for this work bec

---

## Author Comment (AC2) · 18 Apr 2016

**Response to interactive comments on "Speciation of OH reactivity above the canopy of an isoprene-dominated forest" by J. Kaiser et al.**

We thank the reviewers for the thorough reading and helpful comments. Below, we address each remark individually. Original comments are in green, and our response follows, indented and in black. Our pagination refers to the pdf of the discussion paper. Beyond the changes to the manuscript outlined below, please note a change in supplement "entrainment" scenario which corrects a previous error in the calculations for the dilution sensitivity test. These changes have no impact on our analysis or conclusions.

**Referee #1**

(1) The abstract does not contain any quantitative information about the level of agreement between model and measured OH reactivity, which is the main result. There is a long introduction to the abstract and it is not until line 18 that any points specifically relevant to the results of this study are presented. Lines 10-17 need to be moved to the introductory material in the main paper, and some of the main results from the study (overall level of agreement (i.e. that the model underpredicts), slopes of main correlation plots, OH reactivity versus isoprene calculated reactivity, diurnal behaviour, statement that there are large model overprediction of OVOCs with values etc.) need to be stated. The abstract needs to be extended considerably in terms of summarising the main results.

> We have substantially changed the abstract to eliminate much of the background and add a more quantitative and detailed summarization. The new abstract reads:

> *Measurements of OH reactivity, the inverse lifetime of the OH-radical, can provide a top-down estimate of the total amount of reactive carbon in an airmass. Using a comprehensive measurement suite, we examine the measured and modeled OH reactivity above an isoprene-dominated forest in the South East United States during the 2013 Southern Oxidant and Aerosol Study (SOAS) field campaign. Measured and modeled species account for the vast majority of average daytime reactivity (80-95%), and a smaller portion of night-time and early morning reactivity (68-80%). The largest contribution to total reactivity consistently comes from primary biogenic emissions, with isoprene contributing ~60% in the afternoon, ~30-40% at night, and monoterpenes contributing ~15-25% at night. By comparing total reactivity to the reactivity stemming from isoprene alone, we find that ~20% of the discrepancy is temporally related to isoprene reactivity, and an additional constant ~1 s$^{-1}$ offset accounts for the remaining portion. The model typically overestimates measured OVOC concentrations, indicating that unmeasured oxidation products are unlikely to influence measured OH reactivity.*

*Instead, we suggest that unmeasured primary emissions may influence the OH reactivity at this site. While the magnitude of OH reactivity is related to RO2 production and subsequent ozone formation, determining the molecular structure of compounds related to missing reactivity is essential to understanding its impact.*

Paper

(1) Page 7 (there are no page numbers, so pages here refer to pages with the cover page as page 1) lines 2-3. In the measurement of OH reactivity, how was the zero of the instrument determined, and what is the value? Also, presumably in this environment the level of NO is low enough that any corrections for recycling of OH from HO2+NO within the sampling airstream are not necessary? If this is the case it should be stated. It would also be worth stating how the accuracy of the instrument is checked using known OH sinks.

The operating procedures for the OH reactivity instrument are described in Mao et al. (2009). The OH reactivity instrument zero is determined by measuring the wall loss of the OH radical while using a clean carrier gas. The uncertainty in the zero is $0.5$ s$^{-1}$, with $2\ \sigma$ confidence. The recycling of OH from $HO_2$+NO was corrected by taking into account measured $HO_2$ decays. The accuracy of the instrument was verified using gasses with well-known reaction rate coefficients ($C_3F_6$ in the field, and CO, propane, propene, and isoprene in the lab).

In the manuscript, we now state the value of the instrument offset during SOAS and refer the reader to Mao et al. (2009) for a description of the measurement technique.

(2) Page 7, line 25, MCM (not MVM).

This has been corrected.

(3) Page 8, line 29. Would any process immediately regenerate OH? The timescale may be fast compared with the OH decay, but the actual values should be compared, and one shown to be much faster than the other, rather than just this statement.

According to the version of the MCM used here, some processes immediately regenerate OH, most notably the reactions of some hydroperoxides with OH. For example, the product of ISOPOOH + OH reaction is IEPOX + OH. Because loss and production of OH is simultaneous, the ISOPOOH + OH reaction would not impact the measured OH decay. We have clarified this in the manuscript.

(4) Page 9, line 1. Small is subjective, please give a % value here to evidence this.

We now state that the average total calculated reactivity of species that immediately regenerate OH is $0.6 \pm 0.3$ s$^{-1}$.

(5) Although not the subject of this paper (and measured OH is used to constrain the model), it would be useful to state the level of agreement between the model and measured OH and other measured radicals (HO2 and maybe RO2). It would be useful just to confirm how the model performs for these species (given that the model comparison for OVOCs is discussed later).

We cannot provide a comparison of measured and modeled OH or $RO_2$ because our model is constrained to measured OH values and $RO_2$ measurements are not available.

$HO_2$ was not constrained and can be compared to measurements. Our results are shown below. A manuscript comparing measured and modeled OH and $HO_2$ has recently been submitted to J. Atmos. Sci. (Feiner et al. 2016). Slightly different model configurations and selection of days results in a small difference between studies, but both show similar results for $HO_2$. Feiner et al. also find good agreement for OH (chemically zeroed measurement) and model values. We direct readers to the Feiner et al. paper for a more complete discussion of the $HO_X$ budget for this campaign.

[Figure]

**Figure 1.** Measured and model $HO_X$ concentrations. The results of Feiner et al. are shown in (a) and (b). Individual observations are shown in grey, and other points are hourly median values. Our results are shown in (c). As in the figures in the manuscript, lines represent average values, and shaded regions and error bars represent 1 σ standard deviation.

(6) Page 10, line 1. The largest discrepancy is observed at 0700 LT (32%). Can the authors comment on the general shape of Figure 2?

The general shape of Figure 2, including the peak of missing reactivity in the early morning, is addressed in the discussions section (page 8, starting at line 13).

(7) Page 10, line 16. It is stated that the slopes are not much larger than one. This is rather subjective as the observed slope of measured OH reactivity versus that from isoprene itself is 1.43, which although not a factor of 3, 5 etc., is significantly greater than 1 (by almost 50%). This statement ought to be qualified. It can be seen clearly from Fig 3 that at times the reactivity due to isoprene is considerably less than the total measured OH reactivity.

This sentence has been eliminated, as the numbers themselves adequately convey the relationship between total reactivity and isoprene.

(8) Page 12, line 3. Not slope of fig 3? You mean slope of fig 4?

This has been corrected to refer to Figure 4.

(9) It is difficult to compare the temperature dependence observed with that of Di Carlo et al., as the range of values of temperature only seem to have a limited range where they overlap. The parameterisation of Di Carlo seems to stop at 300 K, and in this region the current work's function does not change much. It is only above 300 K that the function for this work becomes significantly steeper than that of Di Carlo. Some further discussion of this is needed, and the parameterisation of Di Carlo needs extending to higher temperatures compare easily.

The parameterization shown in DiCarlo et al. (2004) is based on observations at the lower temperature range shown. Because extrapolating the fit may misrepresent the observations and conclusions of Di Carlo et al., we do not extend their parameterization to higher temperatures. However, we now highlight that the parameterizations are based on different ranges of temperatures.

(10) Page 12. The paper states that ". . .the model is missing ~ 1 s-1 that is temporally unrelated to the oxidation of . . ... (Figure 3)". However, inspection of Figure 3 suggests that the difference between the measured OH reactivity (black points) and the cumulative modelled reactivity is more than 1 s-1? It looks more like between 2 and 5 s-1 (depending on time of day). Some modification of this statement is therefore needed.

While the total missing reactivity is larger than 1 $s^{-1}$, we differentiate between the portion temporally related to isoprene (slope in Figure 4) and the offset (intercept in Figure 4). Increasing the effective reactivity of isoprene by 20% and adding 1 $s^{-1}$ would account for the entirety of the 2-5 $s^{-1}$ of missing reactivity. In our discussion, we address separately the difference in slope and intercept. This has been clarified in the manuscript.

(11) Page 12, line 12, missing "of"

This has been corrected.

(12) Page 13, line 2. Again 1s-1 seems to be a significant underestimate of the difference between measured and modelled shown in Figure 3. The amount of sesquiterpenes would therefore need to be more than ~ 200 ppt.

See response to comment 10.

(13) Page 13, line 22. The values quoted here are not consistent with the values quoted on page 10. The measured slope was quoted as 1.43 there, and modelled slope 1.22. Also, it says +/- 02 (should be 0.02).

We have corrected this error.

(14) Table 1. Glyoxal row, space between 9% (1 s)

This has been corrected.

(15) Figure 1. Please plot this graph from the origin (0,0) as it will be more informative. At present it is rather misleading as it suggests the points go to the origin.

We have remade the figure as suggested.

(16) Figure 1. The slope of the linear least squares fit weighted by uncertainty must be stated in the caption. This is one of the most important results of the paper.

The caption now includes the equation of the linear fit.

(17) Caption for Fig 2, second line, "Points the in gray. . ." needs correction

This has been corrected.

(18) Figure 3. As commented above the difference is considerably more than 1s-1 quoted in the text (seems to be 2-5 s-1 depending on time of day). Suggest reversing the order of the legend. At present the measurement is at the bottom, with Inorganics at the top, whereas the figure is measurement at the top, and Inorganic at the bottom.

See response to comment 10. The figure has been remade as suggested.

(19) Figure 4. Given the text gives the slopes at 1.43 and 1.22 (note inconsistency with page 13), the equation given on the figure needs to reflect this quoted accuracy. What does "weighted by uncertainty" mean?

The figure has been remade as suggested. The uncertainty in OH reactivity measurements (20%) and in the model (20%, based on the uncertainty in isoprene measurements) is noted in the manuscript. References for linear regression given uncorrelated errors are provided.

(20) Figure 6. See comments above about the degree of overlap in T for the solid and dashed lines. What is the value of Greek alpha for each?

See response to comment 9. DiCarlo et al. (2004) provide only a $\beta$ value in the text of their manuscript, and we can therefore not provide an $\alpha$ value. The data shown here was taken from Figure 2 of their manuscript and replotted.

**Referee #2**

The main thrust of the paper is that the modeled reactivity agrees well with the measured reactivity when constrained to measured OVOCs and that primary emissions dominate the OH reactivity at this site, in contrast to previous studies in similar environments. However this conclusion is not highlighted clearly enough in the paper and the authors attempt to extend this conclusion to our understanding of RO2 production and O3 production without providing sufficient modeling studies to support this conclusion. The fact that the model significantly overestimates the measured OVOCs suggests that our understanding of RO2 chemistry and O3 production is still incomplete.

Unfortunately, there is little discussion as to potential reasons why the model significantly overestimates the observed OVOCs, although the paper mentions that uncertainties associated with modeled dilution rates of these species may be responsible. The paper would benefit from an expanded discussion of potential reasons for the model overprediction of OVOCs and their implications.

> Because other manuscripts focus on OVOCs (Su et al., 2015; Xiong et al., 2015) and on the $HO_X$ budget (Feiner et al., 2016), we specifically focus only on OH reactivity given a well constrained model. OH reactivity inherently has implications for $RO_2$ production, which are briefly discussed but not the major focus of this paper. The largest uncertainty influencing OVOC concentrations is dilution, which is discussed in depth in the supplement.

Specific comments:

1) As discussed above, a main conclusion of the paper appears to be that because the modeled OH reactivity (constrained to the OVOCs) agrees well with the measured OH reactivity that the total RO2 production rate and therefore O3 production is well understood (page 9 lines 10-11).

> As stated by the manuscript title, the focus of this work is on the speciation of measured and modeled OH reactivity. The extension to $RO_2$ production (and SOA formation) is an implication that is derived from this work, but it is not the main focus. Our abstract has been rewritten to better reflect our primary focus and summarize our findings.

However, the fact that the model overestimates isoprene oxidation products and other OVOCs suggests that when the model is not constrained to measurements of these compounds that the model may not be able to reproduce total RO2 production, as many of these OVOCs are produced from RO2/HO2 chemistry, such as ISOPOOH and HPALD. The authors need to provide more information to justify this conclusion.

> It is true that when the model is not constrained to measured OVOCs that $RO_2$ production is likely represented less accurately. However, because these measurements are available, we are able to better constrain our model. As a well constrained model can capture OH reactivity, then the essential $RO_2$ production rate (essentially given by the

OH + VOC reaction rate) is likely also captured in this scenario. We do not imply that an unconstrained model would accurately capture the $RO_2$ production rate.

2) One test of the ability of the model to reproduce the observed OH reactivity would be to unconstrain the model to the measured OH and the measured OVOCs. It's not clear from the information given in the paper whether unconstraining the model to the measured OH impacts the modeled OH reactivity as the paper does not state how well the model is able to reproduce the measured OH when constrained to the measured species, or how sensitive the modeled OH reactivity is to the OH concentration. The paper would be stronger if the authors provided several model scenarios to compare to the measured OH reactivity, such as i) a base scenario where the model is constrained only by the traditional measured VOCs, NOx, etc., ii) a scenario where the model is further constrained by the measured OH but not constrained by the measured OVOCs, and iii) the scenario where the model is constrained by all measured species.

One of the primary goals of this manuscript is to determine if there is a large source of missing reactive carbon in our set of measured and modeled species. We therefore focus on the ability of a well-constrained model to capture OH reactivity, and if not, on potential sources of discrepancies. For this reason, we constrain to measurements whenever possible. This includes constraining to measured OH.

A manuscript comparing measured and modeled OH and $HO_2$ has recently been submitted to J. Atmos. Sci. (Feiner et al. 2016). Slightly different model configurations and selection of days results in a small difference between studies, but the results are similar to ours (see below). Because measured and modeled OH and $HO_2$ are in good agreement, and because measured species contribute most substantially to OH reactivity, constraining these radicals has minimal impact on OH reactivity. We direct the reader to the Feiner et al. manuscript for a more complete discussion of radical chemistry.

3) It is surprising that the authors chose to constrain the model to the measured OH but not to the measured $HO_2$, given the importance of peroxy radical chemistry to the formation of OVOCs such as ISOPOOH and HPALD. Were $HO_2$ measurements not available?

We did not constrain our model $HO_2$ measurements due to the higher than typical uncertainty ($\pm 40\%$ at $2\sigma$ confidence level) stemming from the methods used to suppress the $RO_2$ interferences.

Does constraining the model to the measured OH reproduce the measured HO2? Or does this model scenario overestimate HO2 leading to the overestimation of the observed OVOCs? If the model is constrained to both measured OH and HO2, are the modeled OVOCs in better agreement with the measurements? These and other tests of the model would provide important information regarding the reasons for the model's inability to reproduce the observed OVOCs.

Shown below are our model results for OH and $HO_2$ as well as the results those of Feiner et al. (2016). Measured and modeled OH and $HO_2$ are in good agreement and have minimal impact on OVOC concentrations. As other studies have focused on the

measured/modeled agreement of OVOCs, we focus instead on an OVOC-constrained model to examine OH reactivity. However, we now show our results in the supplement.

It is important to note that compared to the effects of constraining/not constraining $HO_X$, the range of dilution scenarios shown in the supplement is by far the largest source of uncertainty in modeled OVOC concentrations. OVOCs are highly sensitive to the assumed magnitude and diurnal variability of the dilution rate, which is typically not well parameterized in 0-D box models.

[Figure]

**Figure 1.** Model results of constraining or calculating OVOCs and $HO_X$ on OH reactivity, $HO_X$, and specified OVOCs. Error bars on the measurement represent 1σ diurnal variability. All scenarios use a constant dilution rate of 4 day$^{-1}$.

[Figure]

**Figure 2.** $HO_X$ budget as studied by Feiner et al. (2016).

**References**

DiCarlo, P., Brune, W. H., Martinez, M., Harder, H., Lesher, R., Ren, X., Thornberry, T., Carroll, M., Young, V., Shepson, P., Riemer, D., Apel, E., and Campbell, C.: Missing OH reactivity in a forest: evidence for unknown reactive biogenic VOCs, Science, 304, 722–725, doi: 10.1126/science.1094392, 2004.

Feiner, P. A., Brune, W. H., Miller, D. O., Zhang, L., Cohen, R. C., Romer, P., Goldstein, A. H., Keutsch, F. N., Skog, K. M., Wennberg, P. O., Nguyen, T. B., Teng, A. P., DeGouw, J. A., Koss., A., Wild, R. J., Brown. S. S., Guenther, A., Edgerton, E. S., Baumann, K., and Fry, J. L.: Testing Atmospheric Oxidation in an Alabama Forest, submitted to J. Atmos. Sci., 2016.

Su, L., Patton, E. G., Vilà-Guerau de Arellano, J., Guenther, A. B., Kaser, L., Yuan, B., Xiong, F., Shepson, P. B., Zhang, L., Miller, D. O., Brune, W. H., Baumann, K., Edgerton, E., Weinheimer, A., and Mak, J. E.: Understanding isoprene photo-oxidation using observations and modelling over a subtropical forest in the Southeast US, Atmos. Chem. Phys. Discuss., 15, 31621-31663, doi:10.5194/acpd-15-31621-2015, 2015.

Xiong, F., McAvey, K. M., Pratt, K. A., Groff, C. J., Hostetler, M. A., Lipton, M. A., Starn, T. K., Seeley, J. V., Bertman, S. B., Teng, A. P., Crounse, J. D., Nguyen, T. B., Wennberg, P. O., Misztal, P. K., Goldstein, A. H., Guenther, A. B., Koss, A. R., Olson, K. F., de Gouw, J. A., Baumann, K., Edgerton, E. S., Feiner, P. A., Zhang, L., Miller, D. O., Brune, W. H., and Shepson, P. B.: Observation of isoprene hydroxynitrates in the southeastern United States and implications for the fate of NOx, Atmos. Chem. Phys., 15, 11257-11272, doi:10.5194/acp-15-11257-2015, 2015.

---

## Author Response (AR2)

**Additional minor revisions to "Speciation of OH reactivity above the canopy of an isoprene-dominated forest"**

We thank the reviewers for the additional comments, and have made the suggested edits. Below, we address each remark individually. Original comments are in green, and our response follows, indented and in black.

1. The details provided on the OH reactivity instrument are still rather brief although there is a reference to Mao et al. (2009) where further details can be found.

>A paragraph describing the OH reactivity measurement is now included:

>*OH reactivity was measured by adding OH to an airstream using a moveable wand, and monitoring the decay of the OH radical by laser-induced fluorescence (Mao et al., 2009). The instrument zero (4.3 $s^{-1}$) is determined by measuring the wall loss of the OH radical while using a clean carrier gas. The uncertainty in the zero is 0.5 $s^{-1}$, with 2 σ confidence. The recycling of OH from $HO_2+NO$ was corrected by taking into account measured $HO_2$ decays. The accuracy of the instrument was verified using gasses with well-known reaction rate coefficients ($C_3F_6$ in the field, and CO, propane, propene, and isoprene in the lab). Further details about the operating procedures for the OH reactivity instrument are described in Mao et al. (2009).*

2. Although in the response the uncertainty in the zero is now stated (0.5 s-1, 2 sigma), the actual zero of the instrument is still not stated, and needs to be as this is a fundamental parameter. Also in the response it is not clear what is meant by "instrument offset"? In the revised manuscript, it states that the "zero" is (0.5 s-1, 2 sigma), whereas should this be the uncertainty in the zero, as stated in the response? This inconsistency needs to be addressed. The zero should be stated with its uncertainty.

>Originally, "offset" and "zero" were used interchangeably. We now consistently refer only to the instrument "zero" (4.3 $s^{-1}$).

>The first revised manuscript incorrectly listed the uncertainty in the zero as the value of the zero. Our new revision states the value of the zero (4.3 $s^{-1}$) and alongside its uncertainty (0.5 $s^{-1}$, 2 σ, see above).

3. Given that uncertainty in the dilution rate is stated in the response to represent the largest source of uncertainty in modelling OVOCs - the amount of material on this point in the main paper ought to be increased (most of the discussion on this important point is in the supplement).

The majority of the discussion on the sensitivity of model OVOC concentrations to dilution rate has been moved from the supplement to the main paper (section 3.2). The key conclusion regarding the minimal effect of dilution rate on OH reactivity is emphasized.

4. I would recommend that the authors add an additional paragraph to the main text describing the additional results illustrated in Figure S5, as there is little discussion of these results in the main text or in the supplement.

We summarize the important findings of Figure S5 in section 2.2:

*Figure S5 provides model results for $HO_X$ ($HO_2$ + OH), OH reactivity, and two first-generation OVOCs given a variety of possible constraints on $HO_X$ and OVOCs. The result essential to this analysis is that model $HO_x$ is in good agreement with measurements and has a minimal impact on calculated OH reactivity and model OVOC concentrations. Our results are in agreement with Feiner et al. (2016), which also employs a 0-D box model and the MCM isoprene oxidation mechanism. Feiner et al. (2016) provide a detailed discussion of the observed and modeled radical budget which is beyond the scope of this manuscript.*

5. I would also recommend deleting the sentence on page 9 line 16 ("This suggests that the total RO2 production rate and resulting O3 formation are likely well understood") as it detracts from the main conclusions of the paper regarding the ability of the model to reproduce the observed total OH reactivity.

In the first two paragraphs of the introduction, we motivate the study of OH reactivity by outlining the connection between OH reactivity and $RO_2$ production rate, which is in turn tied to $O_3$ production. The manuscript then shows the good agreement between measured and modeled OH reactivity. Therefore, we maintain that the above statement highlights a noteworthy consequence of the analysis provided in the manuscript.

[revised manuscript text omitted]

We thank the reviewers for the additional comments, and have made the suggested edits. Below, we address each remark individually. Original comments are in green, and our response follows, indented and in black.

1. The details provided on the OH reactivity instrument are still rather brief although there is a reference to Mao et al. (2009) where further details can be found.

> A paragraph describing the OH reactivity measurement is now included:

> *OH reactivity was measured by adding OH to an airstream using a moveable wand, and monitoring the decay of the OH radical by laser-induced fluorescence (Mao et al., 2009). The instrument zero (4.3 s$^{-1}$) is determined by measuring the wall loss of the OH radical while using a clean carrier gas. The uncertainty in the zero is 0.5 s$^{-1}$, with 2 σ confidence. The recycling of OH from HO$_2$+NO was corrected by taking into account measured HO$_2$ decays. The accuracy of the instrument was verified using gasses with well-known reaction rate coefficients (C$_3$F$_6$ in the field, and CO, propane, propene, and isoprene in the lab). Further details about the operating procedures for the OH reactivity instrument are described in Mao et al. (2009).*

2. Although in the response the uncertainty in the zero is now stated (0.5 s-1, 2 sigma), the actual zero of the instrument is still not stated, and needs to be as this is a fundamental parameter. Also in the response it is not clear what is meant by "instrument offset"? In the revised manuscript, it states that the "zero" is (0.5 s-1, 2 sigma), whereas should this be the uncertainty in the zero, as stated in the response? This inconsistency needs to be addressed. The zero should be stated with its uncertainty.

> Originally, "offset" and "zero" were used interchangeably. We now consistently refer only to the instrument "zero" (4.3 s$^{-1}$).

> The first revised manuscript incorrectly listed the uncertainty in the zero as the value of the zero. Our new revision states the value of the zero (4.3 s$^{-1}$) and alongside its uncertainty (0.5 s$^{-1}$, 2 σ, see above).

3. Given that uncertainty in the dilution rate is stated in the response to represent the largest source of uncertainty in modelling OVOCs - the amount of material on this point in the main paper ought to be increased (most of the discussion on this important point is in the supplement).

The majority of the discussion on the sensitivity of model OVOC concentrations to dilution rate has been moved from the supplement to the main paper (section 3.2). The key conclusion regarding the minimal effect of dilution rate on OH reactivity is emphasized.

4. I would recommend that the authors add an additional paragraph to the main text describing the additional results illustrated in Figure S5, as there is little discussion of these results in the main text or in the supplement.

We summarize the important findings of Figure S5 in section 2.2:

*Figure S5 provides model results for $HO_X$ ($HO_2$ + OH), OH reactivity, and two first-generation OVOCs given a variety of possible constraints on $HO_X$ and OVOCs. The result essential to this analysis is that model $HO_x$ is in good agreement with measurements and has a minimal impact on calculated OH reactivity and model OVOC concentrations. Our results are in agreement with Feiner et al. (2016), which also employs a 0-D box model and the MCM isoprene oxidation mechanism. Feiner et al. (2016) provide a detailed discussion of the observed and modeled radical budget which is beyond the scope of this manuscript.*

5. I would also recommend deleting the sentence on page 9 line 16 ("This suggests that the total RO2 production rate and resulting O3 formation are likely well understood") as it detracts from the main conclusions of the paper regarding the ability of the model to reproduce the observed total OH reactivity.

In the first two paragraphs of the introduction, we motivate the study of OH reactivity by outlining the connection between OH reactivity and $RO_2$ production rate, which is in turn tied to $O_3$ production. The manuscript then shows the good agreement between measured and modeled OH reactivity. Therefore, we maintain that the above statement highlights a noteworthy consequence of the analysis provided in the manuscript.

---

## Author Response (AR3)

**Additional minor revisions to "Speciation of OH reactivity above the canopy of an isoprene-dominated forest"**

Below is the editor's comment in black, with our response in green.

However, I echoed the suggestion raised by one of the reviewers to delete the sentence "This suggests that the total RO2 production rate and resulting O3 formation are likely well understood". Specifically, the agreement between modeled total OH reactivity and measurements can suggest that the RO2 production rate from VOC oxidation is well understand, but this does not necessarily suggest that the total RO2 production rate and thus ozone formation are well understood. The model over-predicted isoprene first-generation oxidation products and it was suggested that the largest uncertainty influencing OVOC concentrations is dilution. Nevertheless, as noted in the manuscript, an extremely high dilution rate would need to be incorporated for dilution alone to account for the difference between model results and measurements. As such, this would seem to suggest that some other factors could be at play and that the fates of radicals (thus ozone production) are not fully understood. I suggest either removing this sentence, or keeping the sentence in the manuscript but adding some caveats to reflect some limitations of this statement. Either way, this would not divert the merit of the manuscript. Once this is addressed, the manuscript can be accepted for publication in ACP.

We have reworked the sentence addressed above to address the concerns of the editor and reviewer. The revised sentence reads:

[revised manuscript text omitted]